# Noncoding RNAs in Vascular Cell Biology and Restenosis

**DOI:** 10.3390/biology12010024

**Published:** 2022-12-22

**Authors:** Denis Efovi, Qingzhong Xiao

**Affiliations:** 1William Harvey Research Institute, Faculty of Medicine and Dentistry, Queen Mary University of London, London EC1M 6BQ, UK; 2Key Laboratory of Cardiovascular Diseases, School of Basic Medical Sciences, Guangzhou Institute of Cardiovascular Disease, The Second Affiliated Hospital, Guangzhou Medical University, Guangzhou 511436, China

**Keywords:** noncoding RNAs, microRNAs, long noncoding RNAs, circRNAs, vascular cells, smooth muscle cells, endothelial cells, restenosis, in-stent restenosis, neointimal hyperplasia, cardiovascular disease

## Abstract

**Simple Summary:**

Angioplasty is a procedure where a stent is inserted to open a blocked blood vessel that is causing issues for a patient. Restenosis is a medical condition that reverses the benefits of angioplasty, and it is caused by injury from the stent, with inflammation, excessive smooth muscle cell growth and the movement of cells to accumulate inside the vessel, as part of a disproportionate healing response to the foreign object. The current treatments for restenosis stop the growth of all cells in the area, which is an issue as endothelial cells are required to keep growing in order to heal the inner layer of the blood vessel, which was damaged by the stent, and prevent issues in the future, such as blood clots. Noncoding RNAs are small pieces of genetic material that are not translated into proteins; however, they are important in controlling different biological processes, some of which are the growth and movement of specific cells involved in restenosis. Therefore, we may be able to target certain noncoding RNAs to only slow down the growth and movement of the type of cell causing the condition, namely smooth muscle cells, while allowing for endothelial cells to keep growing and healing the blood vessel.

**Abstract:**

In-stent restenosis (ISR), characterised by ≥50% re-narrowing of the target vessel, is a common complication following stent implantation and remains a significant challenge to the long-term success of angioplasty procedures. Considering the global burden of cardiovascular diseases, improving angioplasty patient outcomes remains a key priority. Noncoding RNAs (ncRNAs) including microRNA (miRNA), long noncoding RNA (lncRNA) and circular RNA (circRNA) have been extensively implicated in vascular cell biology and ISR through multiple, both distinct and overlapping, mechanisms. Vascular smooth muscle cells, endothelial cells and macrophages constitute the main cell types involved in the multifactorial pathophysiology of ISR. The identification of critical regulators exemplified by ncRNAs in all these cell types and processes makes them an exciting therapeutic target in the field of restenosis. In this review, we will comprehensively explore the potential functions and underlying molecular mechanisms of ncRNAs in vascular cell biology in the context of restenosis, with an in-depth focus on vascular cell dysfunction during restenosis development and progression. We will also discuss the diagnostic biomarker and therapeutic target potential of ncRNAs in ISR. Finally, we will discuss the current shortcomings, challenges, and perspectives toward the clinical application of ncRNAs.

## 1. Introduction

### 1.1. Overview of In-Stent Restenosis

The global healthcare burden of cardiovascular disease (CVD) is evident as it remains the leading cause of death to this day [1]. The majority of CVD is caused by atherosclerosis, a chronic inflammatory condition characterised by the build-up of plaque inside blood vessels leading to lumen occlusion and a reduction in blood flow, causing ischaemia [2]. One of the main therapeutic options in atherosclerosis and associated conditions, such as ST elevated myocardial infarction (STEMI) and angina, is percutaneous coronary intervention (PCI), also known as percutaneous transluminal angioplasty (PTA) when targeting peripheral vessel disease [3]. PCI and PTA are angioplasty procedures that employ the use of a balloon-tipped catheter that is navigated from the peripheral vasculature to the target occluded vessel. The balloon is then inflated to expand the occluded artery and restore blood flow before a stent is implanted to maintain vessel patency in the long term [4]. Alternatively, endarterectomy can be used to treat diffusely diseased vessels by the excision of the atherosclerotic plaque. Initially, an arteriotomy is performed, usually longitudinally, before the atherosclerotic plaque is dissected from the adventitia and extracted, with subsequent patching of the incision site using an arterial or venous graft, most commonly from the internal thoracic artery or saphenous vein [5]. Endarterectomy can be carried out during coronary artery bypass grafting and paired with stent removal to achieve complete revascularisation as a treatment for ISR. Regarding efficacy, both angioplasty with stenting and endarterectomy are similar, as demonstrated by the CREST study [6] when considering death, stroke and MI as outcomes. In symptomatic carotid stenosis patients, endarterectomy was superior to stenting, with lower risk of stroke, death, and MI. However, in asymptomatic stenosis there was no significant difference. As for restenosis, it was more commonly observed following stenting; however, the rates for severe restenosis did not significantly differ [7].

In-stent restenosis (ISR) is one of the main complications following angioplasty, defined as a binary event where there is ≥50% re-narrowing of a blood vessel previously treated by stenting as diagnosed by intravascular ultrasonography or angiography [8]. It is the result of a cascade of molecular and cellular events causing the disruption of vascular physiological homeostasis and neointimal thickening, initiated by localised vascular injury from balloon expansion and stent implantation. Patients with ISR may exhibit a re-occurrence of clinical symptoms, namely unstable angina, and myocardial infarction (MI), which occurs in 5–10% of cases. Therefore, repeat intervention may be required in the form of angioplasty or artery bypass as part of a procedure called target lesion revascularisation (TLR) [9].

The pathophysiology of ISR is multifactorial and involves distinct and overlapping cellular and molecular mechanisms across different vascular cell types (Figure 1). After vascular injury, the pathophysiology of ISR can be subdivided into early, intermediate, and late phases. During the early phase, endothelial denudation occurs following direct mechanical injury, causing the destruction of the intimal endothelial cell layer, followed by inflammation, platelet activation, growth factor and pro-inflammatory cytokine release. In the intermediate and late phases, vascular smooth muscle cells (VSMC) cause neointimal hyperplasia (NIH), a process mediated by phenotype switching caused by inflammation, resulting in the formation of restenosis [10,11,12].

NIH is initiated by vascular injury to the endothelium, causing an inflammatory response that leads to the activation of VSMCs in an attempt to repair the injury. Activation refers to the phenotype switch experienced by VSMCs from a differentiated, contractile, non-migratory, quiescent phenotype to a secretory, synthetic phenotype, the latter of which causes the excessive proliferation and migration of VSMCs from the tunica media into the neointima, paired with the synthesis and deposition of extracellular matrix (ECM) proteins, culminating in the progressive narrowing of the lumen. This process can spontaneously resolve in certain patients; however, excessive progression of NIH supersedes the healing process and causes restenosis requiring TLR. Several factors influence phenotype switching including pro-inflammatory cytokines, such as TNF-α and IL-1β [13], chemoattractant proteins, namely MCP-1 [14], and growth factors including platelet-derived growth factor (PDGF) [15]. These factors induce VSMC phenotype changes by suppressing the expression of smooth-muscle-cell-specific markers that allow for contraction—for example, α-smooth muscle actin (α-SMA), SM22α and SM-myosin heavy chain (SMMHC)—and instead promoting inflammation, ECM synthesis, re-entry into the cell cycle and migration [16].

Endothelial denudation after injury also contributes to restenosis pathophysiology via disruption of the nitric oxide (NO) inhibitory pathway and removal of the nonpermeable intimal layer, followed by long-term endothelial dysfunction after NIH takes place. The loss of endothelium exposes the medial layer, where quiescent VSMCs reside, to pro-inflammatory cytokines, growth factors and chemoattractant proteins, which stimulate migration and proliferation. Additionally, ECs directly inhibit VSMC phenotype switching by the release of NO, which engages the extracellular signal-regulated kinase (ERK) pathway to inactivate RhoA and induce cell-cycle arrest by the upregulation of the cyclin-dependent kinase inhibitor p21^Waf1/Cip1^, leading to the inhibition of cyclin-dependent kinase 2 (cdk2) and halting G_1_/S phase progression. After denudation, this inhibitory process is interrupted, giving way to phenotype switching [17].

### 1.2. Overview of Noncoding RNAs

Noncoding RNAs (ncRNAs) are conventionally described as a class of RNA that is transcribed from the genome but not translated further into a functional peptide. However, recent studies have observed the presence of small open reading frames on some ncRNAs that result in peptide translation [18]. The ENCODE project [19] has revealed that 80% of the human genome is transcribed into RNA, of which only approximately 2% corresponds to protein-coding regions [20]. Contrary to previous beliefs, ncRNAs have been increasingly associated with essential biological processes and signalling pathways, with key implications in regulating gene expression and pathogenesis. The main types of regulatory ncRNAs are microRNA (miRNA), long-noncoding RNA (lncRNA), and circularRNA (circRNA) (Figure 2).

miRNAs are 18–22 nucleotides long with a single-stranded small hairpin structure. The principal pathway of biogenesis begins with genomic DNA transcription, usually of an intronic region, by RNA polymerase II/III [21,22] to form primary (pri)-miRNA, which is then cleaved by the ribonuclease III enzyme ‘Drosha’ and RNA-binding protein ‘DiGeorge Syndrome Critical Region 8′ (DGCR8) to form precursor (pre)-miRNA. The pre-miRNA is shuttled to the cytoplasm via the Exportin5/RanGTP-dependent complex where it is processed by RNase III endonuclease DICER to form mature miRNA. Mature miRNA is associated with the Argonaute (AGO) proteins to form an miRNA-induced silencing complex (miRISC), which can directly bind to the 3′, or 5′ in certain cases, untranslated region (UTR) of target mRNA to induce degradation and prevent translation, thereby controlling gene expression [23]. miRNAs can also localise to the nucleus where they can bind to gene promoter regions to cause transcriptional repression or activation by epigenetic modulation and post-transcriptional gene silencing via slicer-dependent and -independent pathways [24,25]. An example reflecting this is miR-29a and miR-29b, which are localised to the cytoplasm and nucleus, respectively, with the principal difference being the hexanucleotide sequence ‘AGUGUU’ at the 3′ terminus of miR-29b [26].

lncRNAs are >200-nucleotide-long transcripts that do not have significant protein-coding ability. They are subclassified into five main types depending on their genomic locations and transcriptional direction from their nearby coding genes, either as sense, antisense, intergenic, bidirectional or intronic [27]. lncRNAs exhibit great versatility in function, acting in *cis* in the nucleus by modifying chromatin structure and gene expression directly at the locus, or in *trans* in the cytoplasm by interacting with RNA and proteins to affect gene expression independent of the locus [28]. There are more than 50,000 lncRNA transcripts in the human genome according to NONCODE [29], most of which are transcribed from RNA polymerase II and processed similarly to mRNA, resulting in a characteristic 7-methyl guanosine (m^7^G) cap at the 5′ UTR and a poly-adenosine(A) tail at the 3′ UTR. In contrast to mRNA, they are spliced less efficiently and expressed at lower levels [30], with their cellular localisation highly influencing their function in modulating gene expression [31]. lncRNAs have a wide range of functions: they may act as scaffolds to induce chromatin and epigenetic modifications via histone methylation changes; as decoys to cause transcriptional inhibition, sponging, degradation and protein interaction, affecting localisation; as precursor guides for miRNA and an alternative splicing process; and as signals for transcriptional/translational activation from stimuli [32].

circRNAs are circular closed loops of ncRNA that can be exonic, intronic or exonic–intronic [33]. Their unique structure makes them stable and provides additional resistance to degradation by RNase, making them exciting for use as potential diagnostic biomarkers and in therapeutic target applications for cardiovascular diseases such as ISR [34]. It has been reported that circRNAs are generated through multiple validated pathways. In the lariat model, intron removal from pre-mRNA splicing and further back-splicing by covalent joining of the 3′ and 5′ ends results in an intronic circRNA lariat, whereas in some occasions, exon skipping [35] results in circRNAs composed of mixed lariats. In the direct model, RNA-binding proteins (RBP) recognise specific motifs and back-splicing occurs first to form circRNA. The localisation of circRNA to the nucleus is dependent on UAP56 or URH49 [36], where it can recruit proteins to modify chromatin structure or bind to DNA forming an RNA–DNA hybrid, resulting in transcriptional pausing. In the cytoplasm, circRNAs have been observed to manipulate cellular protein structure, act as miRNA sponges to upregulate target mRNA expression and act as protein-coding templates via the internal ribosome entry site [33].

Interestingly, ncRNAs of the same, or a different, type can also interact with each other via miRNA response elements (MREs) to further regulate gene expression. For example, certain lncRNAs and circRNAs act as competitive endogenous RNAs (ceRNAs) to bind with miRNAs and prevent complementary end-target mRNA degradation in a process called ‘sponging’. This ability provides an extensive additional network of regulation, which can be exploited by a therapeutic agent to provide highly precise gene modulation [37]. Increasing evidence supports a critical regulatory role for ncRNAs in vascular cell biology and ISR [38]. This review will explore the role of ncRNAs in regulating the transcription and expression of genes and associated signalling pathways that result in the pathophysiological changes observed in ISR, with the goal of identifying a suitable drug target to hinder ISR formation while addressing current treatment shortfalls.

## 2. miRNA in Vascular Cell Biology and ISR

Numerous miRNAs induce epigenetic modifications and regulate mRNA expression levels, affecting signalling pathways specific to vascular cell functions and NIH during restenosis pathophysiology (Table 1 and Figure 3). Most VSMC-specific contractile marker genes (α-SMA, SMMHC, SM22α, calponin, transgelin), and genes important for proliferation, migration, and apoptosis, contain an evolutionarily conserved CArG box (CC(A/T)_6_GG) in the promoter region [39]. The CArG box and its regulation are the convergence point for many other signalling pathways, under the control of key transcription factors, such as serum response factor (SRF) paired with its muscle-specific co-activator Myocardin (Myocd) [40]. Myocd binds SRF to form an SRF–Myocd ternary complex, which binds to the promoter regions of contractile-specific genes to allow for the transcription and expression of the quiescent, contractile VSMC phenotype [41]. Due to the plastic nature of VSMCs, the phenotype switch to the synthetic state observed in restenosis is dependent on the balance between additional co-repressors, co-activators, and environmental factors. miRNAs are implicated in regulating a large proportion of these signalling pathways and gene targets. Specifically, Krüppel-like factor 4 (KLF4) is often implicated in cardiovascular pathophysiology [42], as an important regulator of VSMC phenotype switching and a target for many miRNAs. KLF4 expression counteracts Myocd and promotes the synthetic VSMC phenotype by preventing SRF association with the contractile-specific gene promoter regions. Moreover, histone deacetylase 2 (HDAC2), which is regulated by miRNAs, causes this transcriptional repression through chromatin condensation and compaction, which blocks CArG site accessibility to SRF [39,43]. PDGF increases the expression of KLF4, while disrupting the SRF–Myocd complex via the phosphorylation and nuclear import of Elk-1, causing Myocd displacement from SRF–Myocd ternary complex and forming an Elk-1–SRF ternary complex that blocks contractile-specific gene expression at the locus [44]. Apart from controlling VSMC functions, emerging evidence also suggests a functional role for miRNAs in EC biology and re-endothelialisation, which provides opportunities for the therapeutic inhibition of ISR while promoting re-endothelialisation after stent implement.

### 2.1. miRNA in VSMCs in ISR

#### 2.1.1. miR-22

miR-22 has been suggested as an attractive target for therapeutic application in coronary atherosclerosis and ISR [89]. miR-22 is an important modulator of the contractile VSMC phenotype, which is downregulated by PDGF during restenosis and conversely upregulated by TGF-β in a p53-dependent manner [51]. miR-22 affects the phenotype switching, proliferation and migration of mature VSMCs during restenosis via four main targets: MECP2, HDAC4, EVI1 [51], and HMGB1 [90]. MECP2 is a strong transcriptional repressor or activator, depending on the molecules it binds to, and is involved in the majority of Rett syndrome cases and neural development by epigenetic regulation [91]. MECP2 was one of the first identified targets of miR-22 [92], with its previous roles in differentiation from embryonic stem cells into VSMCs also supported in mature VSMCs. MECP2 downregulation by miR-22 overexpression prevents H3K9 trimethylation (H3K9me3) at gene promoter regions, thus preventing repression and allowing for contractile-specific gene expression [92]. Similarly, HDAC4 inhibition by miR-22 leads to an increase in the SRF–Myocd chromatin accessibility at the CArG sites and also promotes G_1_-S cell-cycle arrest in a p21- and p27-dependent manner [93]. Finally, EVI1 is the most recent miR-22 target identified, which normally binds the promoter regions of SM22α, αSMA, SRF and Myocd and represses transcription via H3K9me3 enrichment. miR-22 upregulation inhibited MECP2, EVI1 and HDAC4, which prevented synthetic VSMC phenotype switching in vitro, ex vivo and in vivo. The results were further supported in the in vivo mouse femoral artery wire-injury models, which showed that ectopic local miR-22 delivery was able to reverse the synthetic VSMCs back to a contractile phenotype as demonstrated by increased contractile-specific gene expression and a reduced proliferation rate, which also led to a subsequent reduction in neointimal formation. Additionally, this also translated to humans, as results obtained from leg amputation samples with severe neointimal hyperplasia showed increased expression of MECP2 and EVI1 and concurrent miR-22 downregulation in diseased arteries [51]. Consequently, the therapeutic potential of miR-22 in ISR has been further examined in two recent studies using an miR-22-coated balloon [94] and miR-22-loaded Laponite hydrogels [95], respectively, in a rat model of balloon-induced vascular injuries. Most importantly, miR-22-eluting cardiovascular stent has been developed and used to inhibit balloon dilatation-induced ISR in white minipigs [96]. Intriguingly, the authors also observed that miR-22-eluting cardiovascular stent dramatically enhanced VSMC contractile phenotype without interfering EC proliferation, leading to EC growth dominating compared to VSMCs [96]. This emerging evidence has collectively demonstrated that the miR-22-coated stent represents one of the most promising miR-coated stents for reducing ISR and promoting stent re-endothelialisation.

#### 2.1.2. miR-34a

miR-34a is downregulated during ISR pathophysiology and acts to stimulate pro-contractile VSMC phenotype switching by binding to the 3′ UTR of Notch1 [60]. Similarly to miR-22, miR-34a is downregulated by PDGF and significantly upregulated by TGF-β, with roles in the proliferation and migration of VSMCs and implications in stem cell differentiation by the upregulation of Sirtuin-1 (SirT1) [97]. In vitro experiments demonstrated that miR-34a reduced the proliferative and migratory ability of PDGF-treated VSMCs, with no impact on apoptosis. Furthermore, human aortic smooth muscle cells (HASMCs) transfected with miR-34a displayed the same results, showing translation from mice to humans [60]. Notch1 expression is predominantly observed in ECs [98]; however, it is also present in VSMCs following vascular injury and promotes NIH via the activation of the CHF1/Hey2 pathway, which increases susceptibility to growth factors and mediates proliferation via Rho GTPase Rac1 [99]. miR-34a expression inhibits Notch1 signalling and prevents neointimal formation by inhibiting VSMC proliferation and migration, but not affecting apoptosis. Moreover, the roles of miR-34a are cellular-context-dependent, as it has been shown to inhibit SirT1 in ECs to maintain senescence, preventing the proliferation and migration required for re-endothelialisation after vascular injury in other cardiovascular conditions such as atherosclerosis and hypertension [60,100]. Therefore, the precise role of miR-34a in ECs during restenosis needs to be elucidated in future studies, as its effect may vary in a pathophysiology-dependent manner.

#### 2.1.3. miR-124

miR-124 attenuates the proliferation and migration of VSMCs via its downstream targets specificity protein 1 (Sp1), S100 calcium-binding protein A4 (S100A4) and IQ-motif-containing GTPase-activating protein 1 (IQGAP1) [63,64,65]. miR-124 biogenesis is regulated by heterogeneous nuclear ribonucleoprotein A1 (hnRNPA1) through mediation of the processing of primary miR-124 to precursor miR-124 [64]. miR-124 acts to inhibit IQGAP1 mRNA and protein levels when expressed, which leads to a significant reduction in the proliferation and migration of VSMCs [64]. IQGAP1 is specifically involved in the migration of VSMCs via the regulation of actin construction, migration, and cell–cell adhesion by associating with PDGFRβ and facilitating interactions between GTPases and other molecules with focal adhesion assemblies [101]. The induced expression of hnRNPA1, and therefore miR-124, was found to inhibit neointimal formation in femoral wire-injury in vivo models [64]. S100A4 is a calcium-binding protein that acts via the receptor for advanced glycation end products (RAGE) to induce VSMC proliferation and migration, via urokinase-type plasminogen activator and matrix metalloproteinases (MMPs) as well as pro-inflammatory cytokine production via the NF-κB pathway [102]. miR-124 represses the activity of S100A4 to prevent neointimal formation by reducing VSMC proliferation and migration. Sp1 levels were significantly upregulated in vascular injury, paired with decreased SMMHC and αSMA levels, indicating a pro-synthetic VSMC phenotype switch, with miR-124 transfection reversing this, indicating its role in the modulation of VSMC phenotype via Sp1 [65].

#### 2.1.4. miR-143/145

miR-143/145 are separate miRNAs that are co-transcribed by transcription factor SRF and its co-activator Myocd. miR-143/145 were demonstrated to negatively regulate proliferation and promote the contractile phenotype in VSMCs. Specifically, it has been shown that miR-143 inhibits Elk-1 and miR-145 potentiates Myocd transcription while inhibiting CamkIIδ and KLF4. Elk-1 is a ternary complex factor that displaces Myocd from SRF, preventing co-activation, whereas CamkIIδ and KLF4 are factors that prevent SRF binding via chromatin remodelling mediated by HDAC [103]. Moreover, miR-143/145 were observed to suppress the PDGF-mediated proliferation of VSMCs and PKCε-mediated podosome formation and migration [104]. In ISR, miR-143/145 were downregulated by PDGF, with the inhibition of miR-143/145 promoting VSMC proliferation, migration and phenotype switching, whereas induced overexpression resulted in a reversal of VSMCs back to a contractile phenotype with smaller neointimal formation [105], supporting the viability of miR-143/145 in gene therapy in ISR. It is worth noting that VSMCs can communicate and deliver miR-143/145 to vascular ECs via intercellular signalling mediated by the formation of tunnelling nanotubules (TNTs). TGFβ triggers the formation of TNTs, which directly deliver miR-143/145 into ECs and allow binding to their respective targets, hexokinase II and integrin-β8, to regulate angiogenesis. Increased miR-143/145 inhibited EC proliferation and capillary-like structure formation, causing a decreased angiogenic potential [106,107]. Accordingly, we should be cautious when considering miR-143/145 as therapeutic options for ISR.

#### 2.1.5. miR-214-3p

While the majority of neointimal formation in restenosis is attributed to VSMCs migrating from the tunica media of the affected artery, certain studies have shown a contribution of adventitial vascular stem/progenitor cells (AdSPCs) phenotype switching into inflammatory smooth muscle cells (iSMCs), under the regulation of miR-214-3p [82]. iSMCs contribute to ISR by recruiting macrophages to the injury site and producing pro-inflammatory cytokines [108]. AdSPCs were cultured with TGFβ to produce contractile VSMCs, followed by TNFα to induce the iSMC phenotype. Accordingly, miR-214-3p expression was increased by TGFβ but inhibited by TNFα, and the transfection of miR-214-3p upregulated SMC-specific marker gene expression while reducing inflammatory factor expression [82], supporting miR-214-3p’s role in rescuing iSMCs back to a contractile phenotype. Mechanistically, miR-214-3p modulates the sonic-hedgehog-glioma-associated oncogene 1 (Shh-GLI1) axis by inhibiting the Suppressor of Fused (SuFu) and inducing contractile gene expression in SMCs. The SuFu is a negative regulator of SMC marker gene transcription, via the modulation of the nuclear translocation of GLI1, preventing GLI1 from binding to the promoter regions of SRF and α-SMA, thus repressing the transcription of contractile genes. miR-214-3p is complementary to the SuFu via the binding of the 3′ UTR, which inhibits the SuFu and results in SMC contractile gene expression [82]. In the context of VSMC functions, miR-214-3p was closely regulated by different pathogenic stimuli in VSMCs through a transcriptional mechanism. The enforced expression of miR-214-3p in cultured VSMCs and injured arteries inhibited VSMC proliferation and migration through the modulation of NCK-associated protein 1 (NCKAP1) [83], thereby preventing ISR formation and progression.

### 2.2. miRNA in ECs in ISR

#### 2.2.1. miR-126

ECs are important regulators of vascular cell homeostasis after injury. One mechanism by which this occurs is endothelial microparticle (EMP) release following apoptosis and inflammatory activation. EMPs are bioactive molecules that target both ECs and VSMCs, with the role of regulating cell–cell crosstalk. They contain a variety of miRNAs, of which miR-126 is the most highly expressed with roles in EC repair, atherosclerosis, and re-endothelialisation after injury. In ECs, the uptake of miR-126 induced proliferation and migration to promote re-endothelialisation via the inhibition of sprouty-related, EVH1 domain-containing protein 1 (SPRED1), leading to the subsequent activation of the Ras/MAPK pathway to allow for cell-cycle progression and proliferation [66]. Additionally, miR-126 inhibited atherosclerotic plaque formation in a CXCL12/CXCR4-dependent manner [109]. Interestingly, miR-126 exerted an opposite effect on VSMCs, where miR-126 uptake prevented VSMC proliferation and reduced neointimal formation via the inhibition of low-density-lipoprotein-receptor-related protein 6 (LRP6), which in turn affected β-catenin and p21 expression [67]. These studies highlight the potential of using miR-126 as a therapeutic agent with a dual mode of action, inhibiting VSMC proliferation and neointimal hyperplasia, while simultaneously promoting re-endothelialisation and preventing in-stent thrombosis.

#### 2.2.2. miR-200c-3p

Endothelial–mesenchymal transition (EndMT) is a phenomenon that may contribute to the neointimal formation observed in restenosis via an alternative mechanism to medial VSMC proliferation and migration. EndMT is a process by which ECs lose their specific properties, such as angiogenesis and anti-thrombogenicity, instead favouring a mesenchymal-like phenotype with increased proliferation, migration, and ECM production, similar to synthetic VSMCs [110]. miR-200c-3p, a regulator of EC differentiation from stem cells [111], has been identified as a promoter for EndMT switching via interaction with fermitin family member 2 (FERM2) [112]. The inhibition of FERM2 by miR-200c-3p in the cytoplasm allowed for SRF nuclear translocation and the expression of VSMC-contractile markers in ECs. This was supported in vitro in HUVECs with FERM2 knockdown, which exhibited a higher SRF concentration in the nucleus and a higher VSMC-specific marker expression. In contrast, FERM2 functioned to stabilise the EC phenotype via binding with Y-box-binding protein 1 (YBOX1) in the cytoplasm, with the inhibition of FERM2 causing decreased CDH5 mRNA levels, which is an important factor in maintaining EC junction stability. Therefore, miR-200c-3p favours EndMT by upregulation of VSMC contractile markers and the downregulation of EC-specific marker expression in vascular ECs. An in vivo mouse aortic isograft transplantation model was used to confirm the role of miR-200c-3p in promoting neointimal formation via EndMT, and found that the inhibition of miR-200c-3p in grafted arteries prevented EndMT as evidenced by increased expression levels of EC-specific markers (CDH5, PECAM1), but decreased levels of the VSMC-contractile-specific markers transgelin and α-SMA, thereby suppressing neointimal hyperplasia in grafted aorta [112].

On the other hand, miR-200c is downregulated after vascular injury and is part of a feedback loop with KLF4 and SUMO-conjugating enzyme Ubc9, which regulates VSMC proliferation [80]. During vascular injury and subsequent PDGF release, Ubc9 interacts with KLF4 to induce SUMOylation via the JNK and ERK pathways, which results in the recruitment of the co-repressors NCoR, LSD1 and HDAC2 to the promoter regions of p21, which decreases the p21 expression level. Decreased p21 increased VSMC proliferation via G_1_-S phase progression, driving neointimal formation [113]. Interestingly, the SUMOylation of KLF4 provides a possible rationale for its apparent contradicting roles, as it has previously been shown to inhibit VSMC proliferation via the activation of p21 and p27 in other studies [114], while also preventing contractile marker expression [115]. Therefore, protein modifications, such as by Ubc9, may cause KLF4 to exhibit different properties such as contractile marker repression and increased VSMC proliferation after vascular injury [80].

### 2.3. miRNA in Other Vascular Cells in ISR

#### 2.3.1. miR-223

Endothelial denudation during stent implantation exposes the medial layer VSMCs to activated platelets, which release PDGF to mediate phenotypic switching and increased proliferation. Interestingly, one week following stent implantation in rat femoral artery wire-injury models, activated platelets appeared to be internalised by VSMCs. Platelet-derived miR-223 was found to be upregulated in synthetic VSMCs after internalisation, with a subsequent increase in contractile-specific marker expression (α-SMA, SMMHC, SM22α) and a reduction in proliferation and neointima formation. This confirms the role of miR-223 in reversing dedifferentiation and preventing excessive VSMC proliferation via 3′ UTR binding and the inhibition of PDGFRβ, which is required for PDGF activation. The role of miR-223 in rescuing VSMC phenotype and reducing ISR demonstrates the multifactorial nature of the pathophysiology of ISR and the dual roles for platelet-derived miR-223 in vascular injury repair [85].

#### 2.3.2. miR-195

Macrophages (MΦs) are the dominant immune cells in driving the sustained chronic inflammation during restenosis, mainly via the release of pro-inflammatory cytokines and the recruitment of migratory VSMCs. With regards to this, miR-195 is involved in M1-MΦ polarisation and the regulation of IL-1β, IL-6 and TNF-α release, which are key pro-inflammatory cytokines to the induction of VSMC phenotype switching and migration [79]. Polarisation refers to the modulation of M1-MΦs into anti-inflammatory M2-MΦs, with the latter favouring endothelial repair, revascularisation and reducing VSMC migration by the release of TGF-β, VEGF, IL-10, and FGF-2, which is beneficial in restenosis management [116]. miR-195 inhibits Toll-like receptor 2 (TLR2), which is the main initiator of the NF-κB and p38-MAPK-dependent signalling cascades leading to pro-inflammatory cytokine production and M1-MΦ polarisation [117,118]. Additionally, miR-195 was found to be directly implicated in VSMC phenotype modulation by the inhibition of the CCND1/Cdc42 downstream signalling pathway, leading to a synthetic VSMC phenotype and migration. Endogenous miR-195 expression was approximately 80% lower in the balloon-injury rat coronary artery models when compared to no injury, resulting in increased neointima formation [79]. These studies highlight the multi-target therapeutic potential of miR-195, both in favouring the anti-inflammatory M2-MΦ phenotype and inhibiting the Cdc42-mediated phenotype switching of VSMCs to reduce ISR.

## 3. lncRNA in Vascular Cell Biology and ISR

lncRNAs have increasingly become associated with restenosis pathophysiology, although the research effort into miRNA is greater at the current time. lncRNAs display a crucial involvement in the proliferation, migration, and apoptosis of both VSMCs and ECs, with a growing interest in their application in restenosis therapy (Table 2; Figure 4). lncRNAs act via varying mechanisms of action depending on classification (intergenic, sense, antisense, bidirectional, intronic) and subcellular localisation. In the nucleus, they are mainly involved in epigenetic modifications, chromatin remodelling, transcription, and pre-mRNA splicing, whereas in the cytoplasm they impact mRNA translation, stability, and miRNA function. The main mechanisms by which lncRNAs control target gene expression are acting as ceRNAs, scaffolds, decoys, signals, or sponges for miRs [119,120].

### 3.1. lncRNA in VSMCs in ISR

#### 3.1.1. GAS5

lncRNA GAS5 is a well-established tumour suppressor that has been vastly covered in cancer research [145]. Similar to its role in cancer, in vitro results have concluded that the downregulation of GAS5 is associated with the increased proliferation and decreased apoptosis of VSMCs by the activation of the p53 pathway. GAS5 binds and stabilises p53, allowing its transcriptional co-activator, p300, to bind, resulting in the activation and expression of p21 and NOXA, which are responsible for cell-cycle arrest and caspase-3-mediated apoptosis induction, respectively. Adenoviral vector delivery of GAS5 to injured vessels in animal models resulted in decreased neointimal formation [123]. The role of GAS5 in the phenotypic modulation of VSMCs has yet to be covered; however, past studies have demonstrated p53–p300 involvement in maintaining the differentiated, contractile phenotype in VSMCs [146], implying a theoretical pro-contractile phenotype modulation by GAS5. However, this has yet to be experimentally validated. In contrast to its protective role in VSMCs in the context of ISR, the overexpression of GAS5 facilitates M1 macrophage polarisation and promotes inflammation in diabetes [147], atherosclerosis and cancer [148]. Moreover, GAS5 sponges miR-221, leading to the increased production of pro-inflammatory cytokines and MMP-2/9 expression in atherosclerosis [149]. It is important to note that this effect has not been confirmed in restenosis specifically, as the specific pathophysiology and environmental factors in vascular injury may change its role in macrophages.

#### 3.1.2. SMILR

SMILR is an lncRNA located on chromosome 8 that is involved in regulating mitosis and promoting VSMC proliferation when expressed. SMILR upregulates HAS2 expression, a gene located 750 kbp away, which encodes an enzyme necessary for hyaluronic acid (HA) production. The exact mechanism of SMILR in the regulation of HAS2 gene expression is unknown, although Ballantyne et al. [134] suggested it may act as a scaffold or enhancer of the HAS2 promoter region. HA is a critical ECM component secreted by proliferating, synthetic VSMCs and promotes neointimal formation as demonstrated by cuff-injury animal models in Kashima et al.’s study, with downregulation of HA causing a reduction in NIH [150]. Additionally, SMILR expression was upregulated by IL-1/PDGF in VSMCs but not in ECs, which supports its viability as a therapeutic target to halt NIH while maintaining re-endothelialisation. Moreover, SMILR also affects mitosis to increase VSMC proliferation as unveiled by Mahmoud et al.’s [135] study which displayed direct interactions with CENPF, a ubiquitously expressed protein, which may function by kinetochore assembly and adequate chromosome alignment, although the exact mechanism has not been confirmed yet.

#### 3.1.3. MYOSLID

MYOSLID is a VSMC-specific natural antisense lncRNA localised to the cytoplasm involved in regulating the pro-contractile VSMC phenotype and reducing proliferation. MYOSLID is a direct transcriptional target of the SRF/Myocd complex, and it acts in the cytoplasm to regulate TGF-β/SMAD2 phosphorylation and MKL1 nuclear shuttling. Downregulation of MYOSLID results in significantly decreased expression of contractile-specific markers SM22-α, calponin-1 and transgelin, supporting the pro-contractile role. MYOSLID affects contractile-specific gene expression by supporting actin stress fiber formation, a RhoA-dependent actin-cytoskeleton remodelling event in which G-actin fibers are formed into F-actin fibres, leading to detachment and nuclear shuttling of MKL1, an SRF co-activator required for contractile-gene expression [151]. MYOSLID also regulates VSMC differentiation via the TGF-β/SMAD2 axis, functioning in a positive-feedback loop with the phosphorylation of SMAD2, although the exact mechanism is unknown [127].

#### 3.1.4. lncRNA-H19

lncRNA-H19 is an lncRNA that is positively correlated with neointimal formation, which acts as the primary pre-miRNA precursor for miR-675 [152]. Both lncRNA-H19 and miR-675 expression were elevated in vascular balloon-injury rat models, with PTEN being identified as the target of miR-675. PTEN is a tumour suppressor that induces G_1_ cell-cycle arrest via the activation of p21 and p27, paired with the downregulation of cyclin D1/CDK4 and cyclin E/CDK2 complexes [153]. Previous studies have also attributed PTEN expression to increased apoptosis in VSMCs via the PI3K/Akt pathway, although Moon et al. did not confirm this in their study. Overall, lncRNA-H19 functions as a positive regulator of neointimal formation, principally via its enabling of VSMC proliferation by the H19-derived miR-675 inhibition of PTEN.

#### 3.1.5. NEAT1

NEAT1 expression was found to be significantly elevated upon vascular balloon injury. Paired with this, NEAT1 expression was concentrated in the neointima where dedifferentiated VSMCs reside, as well as being found in atherosclerotic plaques in humans. NEAT1 expression is induced by PDGF and contributes to the repression of contractile-specific genes while increasing VSMC proliferation and migration. With regards to phenotype switching, NEAT1 acts via WD Repeat Domain 5 (WDR5), an lncRNA-binding protein that regulates histone 3 lysine 4 methylation (H3K4me3). In vascular injury, PDGF mediates NEAT1 expression, which in turn sequestrates WDR5 from contractile-specific gene loci, causing chromatin condensation mediated by decreased H3K4me3, thus restricting SRF access to the CArG sites, and preventing the binding and transcription of contractile genes [130].

### 3.2. lncRNA in ECs and Macrophages

#### 3.2.1. SENCR

SENCR is an antisense cytoplasmic lncRNA involved in regulating VSMC phenotype and migration ability, as well as proliferation in type II diabetes mice exposed to high glucose levels. The down-regulation of SENCR was associated with decreased contractile-specific markers and Myocd levels, with an associated increase in pro-migratory genes [133]. The exact mechanism is unknown; however, due to its cytoplasmic nature, it is more likely to act on mRNA, proteins or as a miRNA sponge rather than on chromatin to modulate a pro-contractile phenotype. Interestingly, Zou et al. [132] demonstrated that mice with type II diabetes displayed reduced SENCR levels and significantly increased aortic smooth muscle layer thickness, which is relevant due to the increased risk of restenosis as a vascular complication in type II diabetes patients. The overexpression of SENCR in these mice was able to reverse the effects of hyperglycaemia on the excessive proliferation and migration of VSMCs, confirming its role in the modulation of these processes. The mechanism by which this occurs in diabetes is via FoxO1. Specifically, SENCR downregulation in T2DM mice upregulated FoxO1, which bound to the promoter region of TRPC6, resulting in increased VSMC migration and proliferation. However, these results are contradictory to non-diabetic vascular injury cases, where FoxO1 was inactivated by phosphorylation and nuclear exclusion by PDGF, resulting in the increased proliferation of VSMCs [154]. As such, the target of SENCR in non-diabetic vascular injury models remains to be elucidated.

SENCR is also expressed in the nuclei of ECs and is important for commitment in the transition from human embryonic stem cells (hESC) to ECs. SENCR expression was found to increase the proliferation and migration of ECs and upregulate the pro-angiogenic genes CCL5 and CX3CL1, which are required for re-endothelialisation after restenosis [155]. Interestingly, the effect exerted by SENCR upregulation is the opposite in VSMCs, where it inhibits proliferation and migration, making it suitable as a therapeutic target to inhibit NIH while enabling re-endothelialisation. SENCR is also involved in the integrity of the endothelium via its interaction with CKAP4. SENCR binds to the noncanonical RNA-binding domain of CKAP4 and restricts it to the cytoplasm, which in turn prevents the internalisation of CDH5, an important molecule in maintaining EC membrane integrity, permeability, and stable adherent junctions via the CDH5/CTNND1 complex [156]. However, its role and underlying mechanisms in ECs during restenosis have yet to be covered in detail, as the pathophysiology may cause its function to vary when compared to stem cell differentiation.

#### 3.2.2. MALAT1

The role of MALAT1 in regulating the vascular EC phenotype with regards to revascularisation was initially covered by Michalik et al. [126]. They described increased vascular EC migration and sprouting ability, but decreased cell-cycle progression with MALAT1 silencing, which impaired vascularisation both in vitro and in vivo. The silencing of MALAT1 was demonstrated to induce p21 and p27 activity, which contributed to the halt in cell-cycle progression. Since the specific role of MALAT1 was not tested in the context of mechanical vascular injury, but instead in a hypoxic environment, the findings are not directly translatable to restenosis. However, since MALAT1 is already known to affect VSMC phenotype switching [125], its role in EC could be an attractive future study topic.

#### 3.2.3. MEG3

MEG3 negatively regulates angiogenesis and proliferation in vascular ECs via miR-9 sponging. This was only validated in in vitro experiments with HUVECs, which demonstrated reduced capillary-like formation ability and reduced proliferation upon MEG3 overexpression. miR-9 has been reported to positively regulate angiogenesis via the JAK/STAT pathway, thus sponging by MEG3 functions to inhibit this process and reduce angiogenic potential in HUVECs [129]. Additionally, MEG3 levels were found to be inversely correlated with VEGF levels in another study on osteoarthritis [157]. Another study also demonstrated that MEG3 inhibits EC proliferation and migration via the inhibition of miR-21, which targets PTEN, in samples from human CAD samples [158]. These results are not directly translatable to ISR, but offer a first step in understanding the role of MEG3 in ECs after restenosis and specifically re-endothelialisation.

## 4. circRNA in Vascular Cell Biology and ISR

circRNAs are established multifunctional regulators in cancer [159], and they regulate a variety of processes in vascular cell biology including proliferation, migration and apoptosis (Table 3; Figure 5). circRNAs may act as miRNA sponges, directly interact with RNA-binding proteins, induce epigenetic modifications in the nucleus to regulate transcription, or regulate translational expression in the cytoplasm. One of the key mechanisms of action for circRNA is acting as ceRNA to sponge relevant miRs, thus preventing the inhibition and allowing for the expression of the miRs’ target genes. circRNA can also associate with RNA-binding proteins to modulate the cell cycle, for example, in atherosclerosis, where circANRIL binds to Pescadillo ribosomal biogenesis factor 1 (PES1), causing the inhibition of ribosome maturation, which leads to nucleolar stress and p53 expression, with a subsequent decrease in proliferation and an increase in apoptosis [160]. Other mechanisms, such as transcriptional or translational block, are dependent on the localisation of circRNA in the nucleus or the cytoplasm, respectively. The research on the effect of circRNA in restenosis is sparse and mostly in vitro compared to other CVDs such as atherosclerosis, hypertension, and diabetes mellitus; however, these diseases all share the hyperproliferative VSMC and dysfunctional EC characteristics noted in ISR, which gives valuable insights into the general involvement of circRNAs in regulating cell proliferation and migration. Future studies are required to confirm the involvement of certain circRNAs, as the specific restenosis pathophysiology may influence the effect of given circRNAs on cell proliferation and migration [161].

### 4.1. circRNA in VSMCs in ISR

#### 4.1.1. circ-SIRT1

circ-SIRT1 was noted to be downregulated both in vitro, in response to PDGF stimulation, and in vivo as a response to balloon-mediated injury in rat carotid arteries with subsequent NIH, indicating downregulation during restenosis. circ-SIRT1 overexpression in vitro resulted in the increased expression of SM22α and the decreased expression of proliferating cell nuclear antigen (PCNA), supporting its role in reducing proliferation and maintaining a pro-contractile phenotype in VSMCs [168]. circ-SIRT1 directly binds to c-Myc and restrains it to the cytoplasm, which prevents binding to the promoter region and the expression of cyclin B1, a fundamental sub-unit for CDK1-mediated cell proliferation via p27 phosphorylation [173].

#### 4.1.2. circ-MAP3K5

circ-MAP3K5 is expressed in VSMCs and was shown to be significantly downregulated in arteries of patients with coronary heart disease and in femoral wire-injury mouse models exhibiting NIH. circ-MAP3K5 levels were unchanged after four weeks of observation while neointimal formation increased; however, the adenovirus-mediated transfection of circ-MAP3K5 caused a marked decrease in neointimal formation by decreasing VSMC proliferation and migration [170]. Mechanistically, circ-MAP3K5 has two binding sites for its sponging target miR-22-3p, and the overexpression of circ-MAP3K5 significantly reduces miR-22-3p levels in HCASMCs. miR-22-3p mediates VSMC phenotype switching via Ten-eleven-translocation-2 (TET2), a master epigenetic regulator of VSMC differentiation status that directly binds to the CArG regions of SRF/Myocd to increase chromatin accessibility. Indeed, the knockdown of TET2 resulted in significantly increased H3K27me3 and reduced 5-hydroxymethylcytosine (5-hmC) at contractile-specific gene loci [174]. The sponging of miR-22-3p by circ-MAP3K5 prevents TET2 inhibition and promotes a pro-contractile VSMC phenotype with reduced proliferation and migration ability [170].

#### 4.1.3. circ-Diaph3

circ-Diaph3 was found to be elevated in vivo in common carotid artery mouse models with neointimal formation after balloon injury. circ-Diaph3 levels were negatively correlated with Diaph3 protein levels, the latter being positively correlated to contractile-specific VSMC marker expression. Additionally, miR-148a-5p was targeted by circ-Diaph3 acting as ceRNA in the cytoplasm, which reduced the inhibition of IGF-1R expression by miR-148a-5p [165]. This activity permits the promotion of the IGF-1 signalling pathway, which is commonly implicated in CVDs such as atherosclerosis, hypertension and restenosis, in the latter causing increased VSMC proliferation and ECM production [175,176]. The above findings support a role for circ-Diaph3 in mediating pathological VSMC phenotype switching, proliferation and migration in restenosis.

### 4.2. circRNA in ECs and Macrophages in ISR and Other Vascular Diseases

The roles of circRNA specific to ISR have not yet been explored in detail with regards to function in ECs and macrophages; however, there are studies covering their function in other CVDs such as atherosclerosis. circUSP36 attenuated EC proliferation and migration after overexpression in ox-LDL-treated cells to simulate pathological atherosclerosis plaque development. This process was caused by circUSP36 sponging miR-637 and affecting the downstream target WNT4, which contributes to the endothelial dysfunction observed in atherosclerosis [177]. Similarly, circGNAQ acts as an miR-146a-5p sponge in ECs, which results in Polo-like kinase 2 (PLK2) expression, preventing EC senescence. circGNAQ silencing in human umbilical vein endothelial cells triggered EC senescence and decreased cell proliferation and VEGF angiogenesis in atherosclerosis [178]. Angiogenesis is an important factor of re-endothelialisation in restenosis; thus, exploring the role of circRNAs in this process would contribute to future therapeutic applications.

Macrophage polarisation is an important process influencing the development and progression of several CVDs via M1-macrophage-driven inflammation. Recently, circ-Cdyl was identified to be upregulated in abdominal aortic aneurysm (AAA), contributing to the disease by favouring M1-macrophage polarisation via multiple mechanisms. Firstly, circ-Cdyl binds to the transcription factor IRF4 in the cytoplasm, restricting its entry into the nucleus, and preventing the antagonism of the TLR pathway required for anti-inflammatory M2 polarisation [179]. Secondly, circ-Cdyl targets let-7c and acts as a sponge, which promotes C/EBP-δ expression and pro-inflammatory M1 polarisation [180]. Principally, the aforementioned circRNAs all play crucial roles in pathophysiological processes overlapping with restenosis; therefore, they may be of interest for in-depth exploration in future studies.

## 5. Other ncRNAs in ISR and CVD

Small nuclear RNAs (snRNAs) are confined to the nucleus and are essential components of the spliceosome, which affects pre-mRNA splicing. snRNAs associate with proteins to form small nuclear ribonucleoproteins (snRNPs), with the major classes named as U1, U2, U4, U5 and U6 [181]. The effect of snRNA on vascular cell functions has not been researched directly; however, snRNAs can modulate other ncRNAs. As an example, snRNP regulates lncRNA retention to chromatin and localisation to the nucleus, which is correlated to lncRNA function and gene expression [182]. This was demonstrated by Yin et al., who showed that chromatin association with MALAT1 was decreased after U1 snRNP inhibition [183]. Despite the fact that there is not a direct link between U1 snRNP and restenosis, MALAT1 is involved in ISR pathophysiology; therefore, its regulation by snRNA may provide therapeutic benefit, although more studies are required to determine the exact mechanisms of action and the effects in vascular injury models specifically.

Small nucleolar RNAs (snoRNAs) are a highly conserved class of ncRNAs that fall within the length range 60–300 nucleotides and are mainly localised to the nucleoli. They are further subdivided into two functional classes, C/D box snoRNAs, which guide 2′-*O*-ribose methylation to modulate other ncRNAs and RNAs, and H/ACA box snoRNAs, which modulate ncRNAs via pseudouridylation [184]. Similar to snRNAs, snoRNAs associate with proteins to form functional small ribonucleoprotein particles (snoRNPs), with principal roles in rRNA processing and maturation and mRNA splicing [185,186]. The scope of research on snRNAs and snoRNAs in relation to restenosis is nearly absent; however, there have been associations between snoRNAs, specifically from the 14q32 locus, and CVDs. All of the snoRNAs from this locus are ‘orphan snoRNAs’ with no defined RNA target, suggesting noncanonical mechanisms of action. The 14q32 locus encodes other lncRNAs and miRNAs involved in CVD pathophysiology such as the lncRNA MEG3 [187]. However, Nossent et al. [188] found no association between the 14q32 snoRNAs and restenosis, which occurred in 35% of the PAD patients in the study, or atherosclerosis. Interestingly, it has been reported that SNORD113.2 and SNORD114.1 plasma levels are inversely correlated with platelet activation, which is a contributing factor in the inflammatory cascade driving ISR. Overall, snRNAs and snoRNAs have not yet been implicated in ISR, with future studies being required to elucidate their specific mechanisms of action and associations with the disease.

## 6. Biomarker and Therapeutic Applications of ncRNAs

### 6.1. ncRNAs as Biomarkers in ISR

A biomarker can be defined as any measurable substance or process that influences the incidence or progression of a specific disease. With regards to restenosis, circulating ncRNAs, such as miRNA, have gained interest as attractive biomarker options due to their heightened sensitivity to quantification, ease of extraction and additional stability compared to protein-based biomarkers. Quantification methods with their respective advantages and disadvantages include the gold-standard quantitative polymerase chain reaction (qPCR), microarrays and next-generation sequencing platforms. Real-time qPCR has the advantage of being low-cost and requiring low amounts of material at the cost of producing few data points, whereas microarrays can simultaneously cover multiple miRNAs although requiring higher amounts of starting material; however, both techniques are unable to recognise novel miRNA [189].

The determination of whether a biomarker is suitable for diagnostic or prognostic functions is carried out via the receiver operating characteristic (ROC) curve and the associated area under the curve (AUC). These tools measure the specificity and selectivity of potential biomarkers, with ROC and AUC indicating the degree of discrimination achieved by the biomarker between the disease diagnosis/prognosis and the controls, with an ideal score in disease diagnosis/prognosis being over 0.8 [190]. In this aspect, four main miRNAs, miR-21, miR-100, miR-143 and miR-145, were identified as suitable biomarkers for ISR, each with different specificity, sensitivity, and AUC scores [191]. Moreover, miR-93-5p was identified as an independent predictor of ISR, with an AUC of 0.734, but no disclosed sensitivity and specificity [192].

The diagnostic biomarker potential of lncRNAs in coronary restenosis after PCI in CHD patients was systematically validated as part of a meta-analysis by Liu et al. [193]. Since lncRNA expression has been demonstrated to be abnormal and implicated in restenosis development, this study proved that there is suitable evidence across multiple studies to consider lncRNAs as a viable predictor of ISR occurrence based on specificity and selectivity for the disease. Specific examples include ANRIL, which was determined to be an optimal independent biomarker for ISR. Serum samples from 444 patients who received PCI and a subsequent coronary angiography to test for ISR within 36 months were used to establish ANRIL’s biomarker potential. ANRIL levels demonstrated a positive correlation with the incidence of ISR, both with and without adjusting for confounding variables. ROC curves were constructed, and the AUC was evaluated to be 0.749 with a sensitivity of 68.4% and specificity of 75%, which demonstrates that ANRIL expression correlates with a high risk of ISR. Additionally, ANRIL is a potential prognostic factor due to its independent nature as established by the multiple logistic regression considering known risk factors for ISR, such as smoking, hypertension, diabetes, and high LDL [194].

### 6.2. ncRNAs as Therapeutic Targets in ISR

Rapamycin-derived drugs are at the forefront of the second generation of drug-eluting stent (DES) technology; however, their non-selective mechanism of action decreases cell proliferation in all vascular cell types, preventing re-endothelialisation and giving rise to late stent thrombosis events. The lncRNA SENCR was identified as a possible therapeutic target to allow for re-endothelialisation by circumventing the non-selective inhibition of proliferation by rapamycin. Data showed that SENCR overexpression in ECs increased cell proliferation and migration, as well as angiogenesis [195], supporting a role of SENCR in alleviating the re-endothelialisation inhibition burden by rapamycin and its associated derived drugs in managing restenosis. The inclusion of SENCR in DES may be beneficial not only for permitting re-endothelialisation, but also in reducing NIH and restenosis progression due to the pro-contractile, antiproliferative and anti-migratory properties of SENCR expression previously described in VSMCs, although these must first be validated in in vivo models of vascular injury [132,133].

Gene-eluting stents are the latest advancement to address the shortfalls of DES, namely impaired re-endothelialisation, and subsequent stent thrombosis due to the non-selective nature of proliferation inhibition. Gene-eluting stents employ ncRNAs, such as miRNA, to cause the selective inhibition of VMSC proliferation and NIH, while permitting re-endothelialisation. The preferred method is the loading of the target gene onto a polymer coating on an endovascular stent. However, this is not without challenges as the ideal stent coating would have a controllable dosage profile, good binding efficiency, and targeted, sustained release [196]. In addition, certain stent coatings are lipophilic polymer based, which is unsuitable for hydrophilic miRNAs.

#### 6.2.1. miR-22-Eluting Stent

As previously discussed, miR-22 has been identified as a powerful therapeutic for promoting VSMC contractile phenotype and preventing vascular injury-induced NIH. Based on these discoveries, Wang et al. developed a miR-22-eluting endovascular stent with a self-healable encapsulating microporous structure [96]. miR-22 was chosen as the therapeutic target due to its established regulation of VSMC phenotype, proliferation, and migration in restenosis [51]. VSMCs and ECs were individually seeded onto the microporous structure loaded with miR-22, which led to a significant increase in SMC contractile phenotype, without any disruption of endothelial function, supporting a critical role of miR-22 in inhibiting VSMC proliferation while permitting re-endothelialisation in vitro. Moreover, a porcine coronary artery balloon-injury model was used to test the effects in vivo by comparing the miR-22-eluting stent (MES) to bare stents (BMS). Results showed a 77.5% reduction in neointimal thickness at 28 days following stent implantation, with a significant promotion of α-SMA expression and decreased levels of IL6, EVI1, and Col-I in MES versus BMS [96]. Overall, this study proves the application of miR-22 as part of a gene-eluting stent to selectively inhibit VSMC proliferation and NIH, while permitting for re-endothelialisation in ISR.

#### 6.2.2. Anti-miR-21-Eluting Stent

miR-21 expression is increased in ISR, as it is a contributing factor to the disease. This served as the rationale for the development of anti-miR-21-eluting stents. The local method of delivery obtained via stents addresses the off-target effects observed in low-dose and high-dose systemic delivery, which caused impairment of function in the lung, kidney, liver and heart, although systemic anti-miR-21 was effective in reducing neointimal formation both at low and high doses [197]. Rats implanted with balloon injured human internal mammary artery (IMA) were subjected to either the anti-miR-21 stent or BMS. The results demonstrated a significantly reduced neointimal formation after 28 days with the anti-miR-21 stent compared to BMS, with preserved endothelial function and no significant off-target effects or impairment of function found in other major organs [197], providing a proof-of-concept for the anti-miR-21-eluting stent in treating patients with ISR.

#### 6.2.3. Self-Assembled miR-126 Switch Nanoparticles

Self-assembled nanoparticles are an exciting novel therapeutic option against NIH proposed by Lockhart et al. [198]. This involves the delivery of a mRNA payload to halt cell proliferation, regulated by a miRNA switch to prevent off-target expression. The main component of these nanoparticles is the protein p5RHH, with an excellent ability to spontaneously associate with mRNA and form nanoparticles. These nanoparticles are resistant to RNase A, which is an indispensable feature for in vivo delivery to prevent mRNA degradation. The slight positive charge allows for endocytosis against the negatively charged cell membrane, while concurrently not disrupting membrane structure. Specifically, to inhibit restenosis while permitting for re-endothelialisation, p27^Kip1^ mRNA was associated with p5RHH, with an additional 22-nucleotide complementary miR-126-3p sequence attached to the 5′ UTR of p27^Kip1^. Since miR-126-3p is specific to vascular ECs, this ‘switch’ functions to prevent the exogenous overexpression of p27 in cells that are required to keep proliferating in restenosis, namely ECs, to allow for re-endothelialisation. The cell-specific inhibition of the p5RHH–p27–miR-126-3p switch nanoparticles in VSMCs was confirmed in vitro and in vivo. Most importantly, multiple retro-orbital injections of p5RHH–p27–miR-126-3p into the wire-injured arteries caused significantly reduced NIH, paired with successful re-endothelialisation and undetectable off-target effects in the liver and kidney [198]. This study provides clear evidence to support that the p5RHH–p27–miR-126-3p switch nanoparticles are a promising therapy for ISR.

## 7. Future Outlook and Conclusions

There is undeniable potential in the application of ncRNAs as diagnostic and therapeutic targets for the management of restenosis, as well as other CVDs. Several miRNAs, lncRNAs and circRNAs have been implicated in the pathophysiology of restenosis as explored in this review, with supported in vivo models demonstrating that the modulation of ncRNAs can directly impact ISR. Additionally, serum ncRNAs exhibit excellent stability both to natural factors such as pH changes and RNase activity, and to extreme changes such as boiling, freeze–thawing cycles and extended storage [199,200], which makes them suitable for use as non-invasive diagnostic biomarkers.

Due to the selective nature of ncRNAs and their modulation of specific biological pathways, there is enormous potential for the manipulation of specific processes, which addresses the current issues arising from DES. The primary concern is the disruption of re-endothelialisation due to the non-selective antiproliferative nature of the agents used in DES, which leads to dysfunctional endothelium healing and subsequent stent thrombosis. Therapeutic applications such as the self-assembled miRNA-switch nanoparticles offer highly adjustable selectivity in regulating specific processes by leveraging the natural expression of miRNA across different vascular cell types; for example, miR-126 in ECs.

The choice of delivery method for ncRNA therapy against ISR is also an important consideration, due to the potential unwanted off-target effects in other organs, due to the vast expression of ncRNAs across different cell types. Since ncRNAs are essential to many signalling pathways and disease pathophysiology in different organ systems, caution needs to be taken in modulating their levels systematically. Wang et al. demonstrated that the systemic high-dose delivery of anti-miR-21 was effective in reducing ISR, but with the drawback of significant off-target expression and impairment of function in the liver, lungs, heart, and kidney. The ideal solution to this issue is local delivery via stents or scaffolds, which translates into the rationale behind DES. The cytotoxic agents used in DES such as sirolimus would have severe side effects if their mode of delivery was not via stent coating.

The selective inhibition of different signalling pathways is a golden tool to reduce ISR; however, it is not without drawbacks, as the coating, stent polymer choice and ncRNA delivery mechanism remain methodological challenges. As Wang et al. discussed, miRNAs that are overexpressed after vascular injury are more attractive therapeutic targets, due to the ease of producing antisense oligonucleotides to modulate their expression. Delivery methods to restore the expression levels of downregulated miRNAs are more problematic but possible, such as via viral vectors, which have similar disadvantages of gene therapy, requiring a certain level of infectivity, with nuclear uptake and transcription for precursors. Multiple delivery systems are used in miRNA-based therapy in cancer, including exosomes, polymers, lipid nanoparticles, liposomes, and viruses, which may provide suitable translation into restenosis [201]. For example, anti-miR-21 delivery by nanoparticles against glioblastoma, or exosome-mimetic nanoplatforms for miR-145 delivery against lung cancer [202,203].

Another challenge arises from the unwanted effects of ncRNA expression, namely angiogenesis and proliferation, which are important processes to be upregulated for re-endothelialisation, but in excess, are also important contributors for tumourigenesis. In fact, many ncRNAs important for restenosis are also involved in regulating cancer development and progression. For example, MALAT1 is an oncogenic lncRNA implicated in lung cancer [204], miR-143/145 are downregulated in colorectal cancer [205] and miR-24 acts as a tumour suppressor against gastric cancer [206]. Therefore, since ncRNAs play important roles in regulating cell-cycle checkpoints and tumour angiogenesis, care needs to be taken and long-term safety should be considered when using them as therapies for ISR.

NcRNA-eluting stents seem to be a promising successor to address the drawbacks of current-generation DES. Compared to current antiproliferative agents, they have the advantage of targeting individual processes that contribute to restenosis across multiple vascular cell types. For example, a theoretical ncRNA-eluting stent could be constructed from a polymer scaffold backbone with a biodegradable poly(D,L-lactic-co-glycolic acid) (PLGA) coating [207] containing microspheres with miR-126 to promote EC proliferation for re-endothelialisation, miR-22 to prevent VSMC-driven NIH, and miR-195 to prevent macrophage polarisation and inflammation, thereby controlling ISR formation and promoting endothelium healing.

From a clinical perspective, further studies are required to establish the side effect profile and long-term safety and efficacy of using ncRNA as an ISR therapy. Since most in vivo models are murine, the effects may or may not translate into clinical practice for humans. In the future, larger animal models may be suitable for testing if safety can be guaranteed before human applications begin. Nonetheless, ncRNAs provide an exciting opportunity as diagnostic biomarkers and as therapeutic targets against ISR, with the ability to halt NIH while maintaining re-endothelialisation and preventing stent thrombosis. However, further studies need to be carried out to determine optimal delivery methods, dosage, safety, and off-target effects in humans before translation into clinical practice.

## Figures and Tables

**Figure 1 biology-12-00024-f001:**
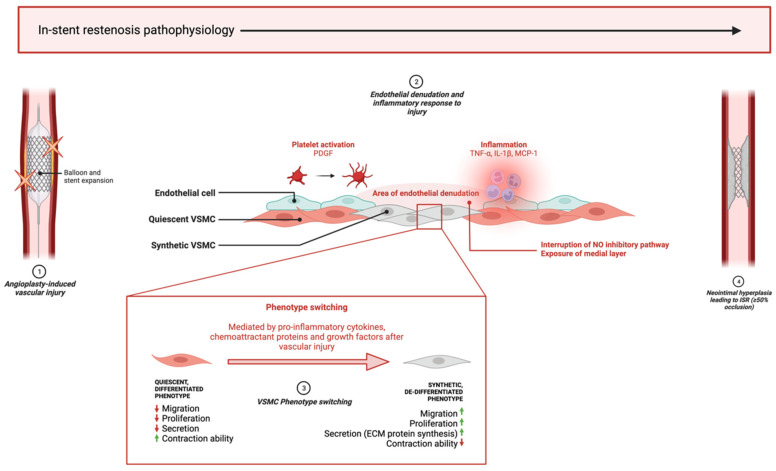
**Key pathophysiological processes of in-stent restenosis**. (1) Mechanical stress from balloon expansion and stent implantation causes vascular injury leading to endothelial denudation and an inflammatory reaction. (2) Inflammatory cells and platelets reach injury site and release pro-inflammatory cytokines, growth factors and chemoattractant proteins which begins the cascade of biological events leading to restenosis pathophysiology. (3) Quiescent VSMCs in the medial layer are exposed by the absent endothelial layer and are modulated to a de-differentiated phenotype that promotes excessive proliferation, migration, and ECM component synthesis in the neointima. (4) ISR develops when neointimal hyperplasia leads to ≥50% lumen occlusion. ‘↑’ and ‘↓’, indicate ‘increase’ and ‘decrease’, respectively. This diagram was created with Biorender.com.

**Figure 2 biology-12-00024-f002:**
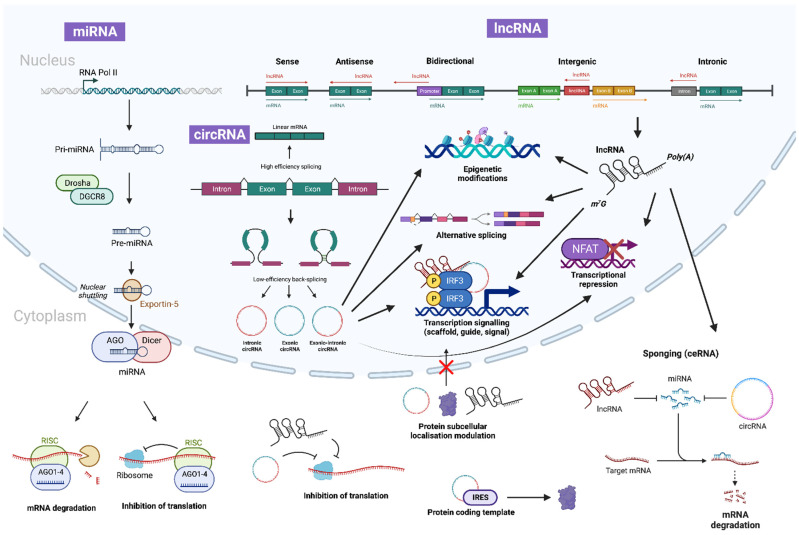
**Biogenesis and canonical functions of ncRNAs.** miRNA begins biogenesis via transcription by RNA polymerase II (or III in certain cases), to form primary miRNA (pri-miRNA), which is then cleaved by Drosha and DGCR8 to form precursor miRNA (pre-miRNA) and exported out of the nuclear pores via the Exportin-5/RanGTP complex. In the cytoplasm it is further processed by AGO and DICER to form a miRNA duplex. The miRNA:miRNA duplex then unwinds to produce mature miRNA that assembles into miRISC alongside AGO. A single mature miRNA can then target multiple mRNAs to induce slicer-dependent or -independent silencing and degradation. Similarly, lncRNAs are transcribed by polymerase II (Pol II) and can be polyadenylated, spliced and 5′-capped. They function as key gene modulators by directly interacting with DNA, RNA and proteins through base pairing or through functional domains. On the contrary, circRNAs are generally generated via back-splicing or exon skipping of pre-mRNAs. Importantly, both lncRNAs and circRNAs function as miRNA sponges, thereby de-suppressing miRNA target gene expression. This diagram was created with Biorender.com.

**Figure 3 biology-12-00024-f003:**
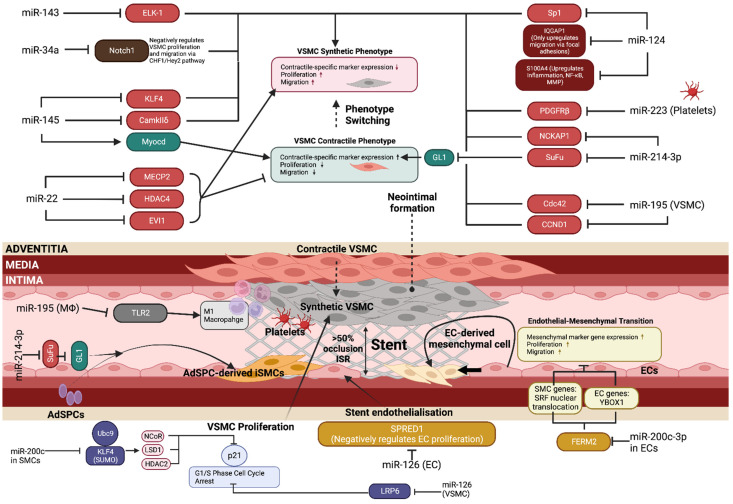
**miRNAs in vascular cell biology and ISR.** The intricate network of biological pathways contributing to restenosis development is highly dependent on miRNA regulation. Multiple vascular cell types (VSMCs, ECs, macrophages, platelets) are involved in the pathological narrowing of the lumen after stent implantation. Most current studies focus on VSMC phenotype switching, proliferation and migration due to this being the major factor driving neointimal hyperplasia; however, it is important to focus on miRNA roles in endothelial cell function, as this can be useful for developing therapeutic agents that inhibit VSMC proliferation while allowing for EC proliferation. EC proliferation and migration is essential for re-endothelialisation and complete vessel healing to prevent thrombosis, a step that is the principal failure of current treatment options in restenosis. ‘↑’ and ‘↓’, indicate ‘increase’ and ‘decrease’, respectively. This diagram was created with Biorender.com.

**Figure 4 biology-12-00024-f004:**
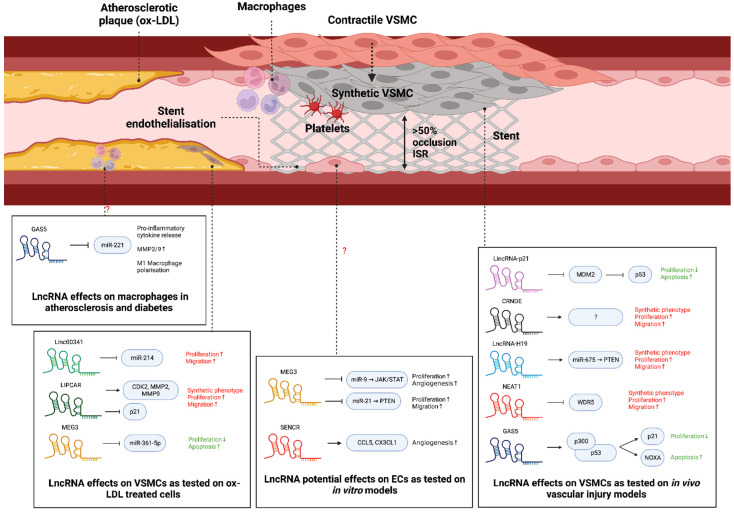
**lncRNAs in vascular cell biology and ISR.** lncRNAs are increasingly being associated with regulating key processes behind restenosis. Specifically, handful of lncRNAs (e.g., lincRNA-p21, CRNDE, H19, NEAT1, GAS5) have been reported to play a role in VSMC functions and ISR formation, while two lncRNAs (MEG3 and SENCR) regulate EC function and possibly stent endothelialisation. Additionally, multiple lncRNAs were found to regulate OX-LDL-induced VSMC proliferation, migration and apoptosis (linc00341, LIPCAR, MEG3) and macrophage polarisation (GAS5), implying a role in atherosclerosis development. ‘↑’ and ‘↓’, indicate ‘increase’ and ‘decrease’, respectively. This diagram was created with Biorender.com.

**Figure 5 biology-12-00024-f005:**
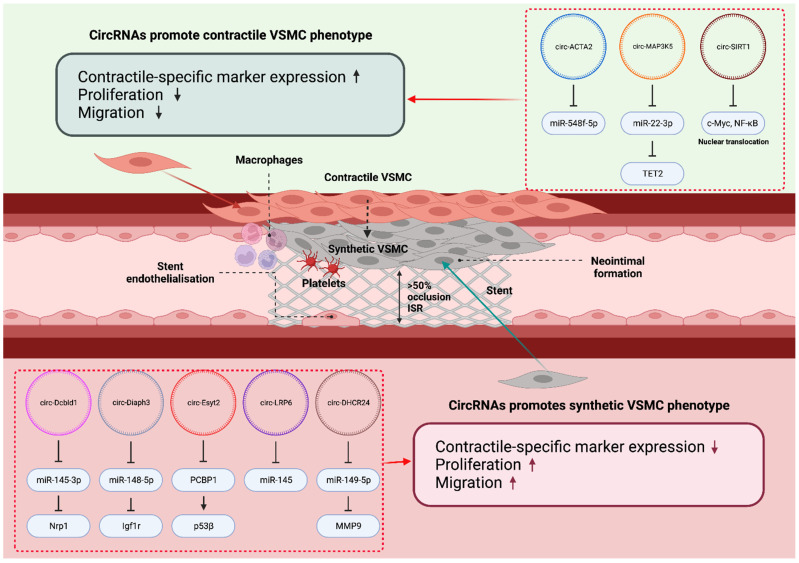
**circRNAs in vascular cell biology and ISR.** A handful of circRNAs were reported to modulate VSMC phenotype switching, thereby promoting or inhibiting ISR formation. ‘↑’ and ‘↓’, indicate ‘increase’ and ‘decrease’, respectively. This diagram was created with Biorender.com.

**Table 1 biology-12-00024-t001:** **miRNAs in vascular cell biology and ISR**. N/A, not applicable; ‘↑’ and ‘↓’, indicate ‘increase’ and ‘decrease’, respectively.

MicroRNA Name	Target and/or Pathway	Vascular Cell Expressed in	Expression Level after Vascular Injury	Vascular Biology & Functions	Animal Model and/or Patient	Effect on ISR	Reference
let-7a	c-Myc, KRAS	VSMC	↓	Proliferation (↓), migration (↓)	Vein graft	↓	[45]
miR-18a-5p	AKT/ERK signalling, Syndecan4, Smad2	VSMC, EC	↑	Pro-synthetic: Proliferation (↑), migration (↑); VSMC differentiation	Carotid balloon injury; patients with in-stent restenosis	↑	[46,47]
miR-21	PTEN, Bcl-2, PDCD4	VSMC, EC	↑	Pro-synthetic (PDGF) Proliferation (↑), migration (↑), apoptosis (↓)	Rat carotid artery balloon injury; human saphenous vein cells post-CABG	↑	[48,49,50]
miR-22	EVI1, HDAC4, MECP2	VSMC	↓	Pro-contractile (PDGF, TGF-β) Proliferation (↓), migration (↓)	Wire/balloon-induced vascular injuries	↓	[51]
miR-23b	Smad3, FoxO4	VSMC, EC	↓	Pro-contractile (TGF-β) Proliferation (↓), migration (↓)	Rat carotid artery balloon injury	↓	[52]
miR-24	TRP-3, Bcl-2	VSMC, EC	↑	Pro-synthetic (PDGF) Proliferation (↑), apoptosis (↑),	Patients with CHD; human primary PASMCs	N/A	[53,54]
miR-26a	Smad1	VSMC, EC	↑	Pro-synthetic (PDGF) Proliferation (↑), migration (↑), apoptosis (↓)	Rat carotid artery balloon injury	↑	[55]
miR-29b	MCL-1, MMP2	VSMC	↓	Pro-contractile Proliferation (↓), migration (↓)	Rat carotid artery balloon injury	↓	[56]
miR-31	MFN2, CREG	VSMC	↑	Pro-synthetic Proliferation (↑), migration (↑)	Rat carotid artery balloon injury; human VSMCs post-CABG	↑	[57,58]
miR-34a	Notch1	VSMC	↓	Pro-contractile Proliferation (↓), migration (↓)	Mouse femoral artery wire denudation injury	↓	[59,60]
miR-92	KLF4, Smurf1	VSMC, EC	↑	Pro-synthetic (PDGF) Proliferation (↑), migration (↑)	Patients with lower limb artery occlusion intervention	N/A	[61]
miR-93	MFN2, Raf-ERK1/2	VSMC	↑	Proliferation (↑), migration (↑)	Rat carotid artery balloon injury	↑	[62]
miR-124	S100A4, IQGAP1, Sp1	VSMC, EC	↓	Pro-contractile Proliferation (↓), migration (↓)	Rat carotid artery wire injury	↓	[63,64,65]
miR-126	SPRED1 (EC), LRP6 (VSMC)	EC, also targets VSMCs	↓	EC—Proliferation (↑), migration (↑) VSMC—Proliferation (↓), migration (↓)	Mouse carotid wire/balloon injury	↓	[66,67]
miR-128	KLF4	VSMC	↓	Pro-contractile (PDGF) Proliferation (↓), migration (↓)	Mouse carotid artery perivascular collar stenosis; patients with atherectomy of the popliteal artery	↓	[68]
miR-132-3p	PTEN, ERK1/2	VSMC	↓	Pro-contractile Proliferation (↓), migration (↓)	Rat PAH model; human PASMCs	↓	[69]
miR-133	Sp1, SRF	VSMC	↓	Pro-contractile (PDGF) Proliferation (↓), migration (↓)	Rat right carotid artery balloon injury	↓	[70,71]
miR-137	IGFBP-5, STAT3	VSMC	↓	Proliferation (↓), migration (↓)	Human aortic smooth muscle cells	N/A	[72]
miR-140-3p	c-Myb, Bcl-2	VSMC	↓	Proliferation (↓), migration (↓), apoptosis (↑)	Rat carotid balloon injury; patients with PAD ISR who accepted limb amputation	↓	[73]
miR-142-3p	DOCK6	VSMC	↓	Pro-contractile (TGF-β) Migration (↓)	Human primary pulmonary artery smooth muscle cells	N/A	[74]
miR-143/145	KLF4, Myocardin, Elk-1, CamkII-δ, PKCε	VSMC, Platelets	↓	Pro-contractile (PDGF) Proliferation (↓), migration (↓)	Mouse left carotid artery ligation injury	↓	[75,76]
miR-146a	KLF4	VSMC	↑	Proliferation (↑), migration (↑)	Mouse vascular wire injury; rat carotid balloon-injury model	↑	[77,78]
miR-195	Cdc42, CCND1, FGF1	Macrophage, VSMC, EC	↓	Pro-contractile Proliferation (↓), migration (↓)	Rat carotid artery balloon injury; human aorta smooth muscle cells	↓	[79]
miR-200c	KLF4, Ubc9	VSMC, EC	↓	Pro-contractile (PDGF) Proliferation (↓)	Rat carotid artery balloon injury	↓	[80]
miR-204	CAV1	VSMC	↑	Proliferation (↑), migration (↑)	Rat right carotid artery balloon injury	↑	[81]
miR-208	p21	VSMC	↑	Proliferation (insulin) (↑)	Vascular smooth muscle cell culture (unknown origin)	N/A	[71]
miR-214-3p	SMYD5, SuFu, NCKAP1	VSMC, AdVSPC	↓	Pro-contractile (non-inflammatory) Proliferation (↓), migration (↓)	Mouse femoral artery wire injury	↓	[82,83]
miR-221/222	p27, p57, PTEN	VSMC, EC	↑	Pro-synthetic Proliferation (↑), migration (↑)	Rat carotid artery balloon injury	↑	[84]
miR-223	PDGFRβ	Platelet-derived, targets VSMCs, ECs	↓ Initially ↑ After	Pro-contractile (PDGF) Proliferation (↓)	Mouse femoral artery wire injury	↓	[85]
miR-638	NOR1	VSMC	↓	Proliferation (↓), migration (↓)	Human aortic smooth muscle cells	N/A	[86]
miR-663	JunB, MYL9, MMP	VSMC	↓	Pro-contractile Proliferation (↓), migration (↓)	Mouse carotid artery ligation injury; Human aortic vascular smooth muscle cells	↓	[87]
miR-1298	Cx43	VSMC	↓	Pro-contractile Proliferation (↓), migration (↓)	Rat carotid artery balloon injury	↓	[88]

**Table 2 biology-12-00024-t002:** **lncRNAs in vascular function and ISR**. ‘↑’ and ‘↓’, indicate ‘increase’ and ‘decrease’, respectively.

lncRNA Name	Target and/or Pathway	Vascular Cell Expressed in	Vascular Biology & Functions	Animal Model and/or Patient	Effect on ISR	Reference
CRNDE	Unknown	VSMC, EC	Pro-synthetic (PDGF) Proliferation (↑), migration (↑)	Rat common carotid artery balloon injury	↑	[121,122]
GAS5	p53	VSMC, EC	Proliferation (↓), apoptosis (↑)	Rat carotid artery balloon injury	↓	[123]
LIPCAR	p21, CDK2, MMP2, MMP9	VSMC	Pro-synthetic (PDGF) Proliferation (↑), migration (↑)	ox-LDL treated human aortic vascular smooth muscle cells	Not measured	[124]
MALAT1	miR-142-3p	VSMC, EC	Pro-synthetic (PDGF) Proliferation (↑), migration (↑)	Human aortic vascular smooth muscle cells	Not measured	[125,126]
MYOSLID	Smad2, MKL1	VSMC	Pro-contractile (TGF-β) Proliferation (↓), migration (↓)	Human coronary artery vascular smooth muscle cells	Not measured	[127]
MEG3	miR-9, miR-21, ABCA1 (via miR-361-5p)	VSMC, EC	VSMC: Proliferation (↓), apoptosis (↑) EC: Proliferation (↓), angiogenesis (↓)	ox-LDL treated vascular smooth muscle cells; human umbilical vein endothelial cells	Not measured	[128,129]
NEAT1	WDR5	VSMC, EC, Macrophage	Pro-synthetic (PDGF) Proliferation (↑), migration (↑)	Rat carotid artery balloon angioplasty	↑	[130]
POU3F3	KLF4/miR-449a	VSMC	Pro-synthetic (PDGF) Proliferation (↑), migration (↑)	Primary human vascular smooth muscle cells; serum samples from patients receiving PCI	Not measured	[131]
SENCR	FoxO1	VSMC, EC	Pro-contractile Proliferation (high glucose conditions) (↓), migration (↓)	Human coronary artery smooth muscle cells; db/db mice exposed to high glucose	Not measured	[132,133]
SMILR	HAS2, CENPF	VSMC	Proliferation (↑)	Human saphenous vein samples after CABG; Human coronary artery smooth muscle cells	Not measured	[134,135]
UCA1	miR-582-5p, hnRNP I	VSMC	Proliferation (hypoxia and high glucose conditions) (↑), migration (high glucose conditions) (↑), apoptosis (hypoxia) (↓)	Hypoxic human pulmonary artery smooth muscle cells; primary human vascular smooth muscle cells	Not measured	[136,137]
linc00341	miR-214/FoxO4	VSMC	Proliferation (↑), migration (↑)	ox-LDL treated primary human vascular smooth muscle cells	Not measured	[138]
lncRNA-Ang362	miR-221/222	VSMC	Proliferation (↑), migration (↑), apoptosis (↓)	Lung tissue samples from patients with PAH	Not measured	[139]
lncRNA-H19	miR-675, PTEN	VSMC, EC	Pro-synthetic Proliferation (↑), migration (↑)	Rat carotid artery balloon injury	↑	[140,141]
lincRNA-p21	p53, MDM2	VSMC	Proliferation (↓), apoptosis (↑)	Rat common carotid artery wire injury	↓	[142,143]
FOXC2-AS1	FOXC2/Notch	VSMC	Pro-synthetic Proliferation (↑), migration (↑)	Human great saphenous vein smooth muscle cells	Not measured	[144]

**Table 3 biology-12-00024-t003:** **circRNAs vascular cell phenotypes and ISR**. ‘↑’ and ‘↓’, indicate ‘increase’ and ‘decrease’, respectively.

circRNA Name	Target and/or Pathway	Vascular Cell Expressed in	Expression Level after Vascular Injury	Vascular Biology & Functions	Animal Model and/or Patient	Reference
circ-ACTA2	miR-548f-5p	VSMC	N/A	Pro-contractile	Mouse femoral artery wire injury	[162]
circ-Dcbld1	miR-145-3p, Nrp1	VSMC	↑	Pro-synthetic, Migration (↑)	Rat common carotid artery balloon injury	[163]
circ-DHCR24	miR-149-5p, MMP9	VSMC	↑	Pro-synthetic, Proliferation (↑), Migration (↑)	Human aorta smooth muscle cells	[164]
circ-Diaph3	Igf1r, miR-148a-5p	VSMC	↑	Pro-synthetic, Proliferation (↑), Migration (↑)	Rat common carotid artery balloon-injury	[165]
circ-Esyt2	PCBP1, p53	VSMC, EC, Macrophages	↑	Pro-synthetic, Proliferation (↑), Migration (↑) Apoptosis (↓)	Mouse carotid artery wire injury	[166]
cIrc-LRP6	miR-145	VSMC	=	Pro-synthetic, Proliferation (↑), Migration (↑)	ApoE KO mice with carotid artery perivascular collar stenosis	[167]
circ-Sirt1	c-Myc, NF-κB	VSMC, EC	↓	Pro-contractile, Proliferation (↓)	Rat carotid artery balloon-injury	[168]
circ-UVRAG	NOVA1	VSMC	N/A	Migration (↑)	Rat vein graft	[169]
circ-MAP3K5	miR-22-3p, TET2	VSMC	↓	Pro-contractile, Proliferation (↓), Migration (↓)	Mouse femoral artery wire injury; human coronary artery smooth muscle cells	[170]
circ-WDR77	miR-124	VSMC	N/A	Proliferation (↑), Migration (↑)	Human vascular smooth muscle cells	[171]
circ-TET3	miR-351-5p	VSMC	↑	Migration (↑)	Rat vein graft	[172]

## Data Availability

Not applicable.

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
