# Peer review of "Noncoding RNAs in Vascular Cell Biology and Restenosis"

_biology, 2022, doi:10.3390/biology12010024_

Round 1

Reviewer 1 Report

I have found the review article a very nice summary of the major ncRNAs involved in the context of ISR.

In the introduction I would also describe the endarterectomy procedure as an alternative to carotid stenting. 

Author Response

Response: Thank you very much for your positive comment about our review article and constructive suggestions about the endarterectomy procedure. As suggested, we have added a short paragraph to briefly discuss using the endarterectomy procedure as an alternative to carotid stenting (Page 3, 1st paragraph, line 12-24). 

Reviewer 2 Report

For the readers, this is a very nice and interesting review. I don’t have any questions to ask about this review, and I would recommend that the editor accept it as is.

Author Response

Response: Your excellent suggestion about our whole manuscript was hugely appreciated. Thank you! 

Reviewer 3 Report

In the manuscript by Efovi and Xiao, the authors comprehensively summarized current studies of ncRNAs in vascular cell biology in the context of restenosis and discussed potential biomarkers and therapeutic applications of ncRNAs. This review also discussed current shortcomings, challenges, and perspectives toward clinical applications of ncRNAs. The manuscript is well-written, although there remain some minor comments as follows.

Minor:

1. Are there any studies about snRNAs and snoRNAs in vascular cell biology and restenosis?

2. When showing examples, the authors should succinctly summarize the major related points, but not list too many details of a study, for example, 5.2.1, 5.2.2, and 5.2.3 can be more succinct.

3. Use “ncRNA” instead of “NcRNA” if it is not at the beginning of a sentence.

Author Response

In the manuscript by Efovi and Xiao, the authors comprehensively summarized current studies of ncRNAs in vascular cell biology in the context of restenosis and discussed potential biomarkers and therapeutic applications of ncRNAs. This review also discussed current shortcomings, challenges, and perspectives toward clinical applications of ncRNAs. The manuscript is well-written, although there remain some minor comments as follows. 

 Response: Thank you very much for your nice comments about our work. 

  1. Are there any studies about snRNAs and snoRNAs in vascular cell biology and restenosis?

Response: Research evidence in this particular area is scarce in the literature. Nonetheless, we have now included one chapter entitled to ‘Other ncRNAs in ISR and CVD’ to briefly discuss a possible link between snRNAs/snoRNAs and restenosis (Page 23/24, two paragraphs).

  1. When showing examples, the authors should succinctly summarize the major related points, but not list too many details of a study, for example, 5.2.1, 5.2.2, and 5.2.3 can be more succinct.

 Response: Thank you very much for your constructive suggestions about these three sections. We have now revised them, making them more concise (Page 25/26; Section 6.2.1 to 6.2.3).

 3. Use “ncRNA” instead of “NcRNA” if it is not at the beginning of a sentence.

Response: Thank you for critical reading our manuscript. We have carefully checked all the text and modified them as appropriate in the revised manuscript.